# C-GAIL: Stabilizing Generative Adversarial Imitation Learning with Control Theory

**Tianjiao Luo[1], Tim Pearce[2], Huayu Chen[1], Jianfei Chen[1], Jun Zhu[1]***

[1]Dept. of Comp. Sci. and Tech., Institute for AI, Tsinghua-Bosch Joint ML Center,
THBI Lab, BNRist Center, Tsinghua University, Beijing 100084, China
[2]Microsoft Research
{luotj21, chenhuay21}@mails.tsinghua.edu.cn
{jianfeic, dcszj}@tsinghua.edu.cn

## Abstract

Generative Adversarial Imitation Learning (GAIL) provides a promising approach to training a generative policy to imitate a demonstrator. It uses on-policy Reinforcement Learning (RL) to optimize a reward signal derived from an adversarial discriminator. However, optimizing GAIL is difficult in practise, with the training loss oscillating during training, slowing convergence. This optimization instability can prevent GAIL from finding a good policy, harming its final performance. In this paper, we study GAIL's optimization from a control-theoretic perspective. We show that GAIL cannot converge to the desired equilibrium. In response, we analyze the training dynamics of GAIL in function space and design a novel controller that not only pushes GAIL to the desired equilibrium but also achieves *asymptotic stability* in a simplified "one-step" setting. Going from theory to practice, we propose Controlled-GAIL (C-GAIL), which adds a differentiable regularization term on the GAIL objective to stabilize training. Empirically, the C-GAIL regularizer improves the training of various existing GAIL methods, including the popular GAIL-DAC, by speeding up the convergence, reducing the range of oscillation, and matching the expert distribution more closely.

## 1 Introduction

Generative Adversarial Imitation Learning (GAIL) [1] aims to learn a decision-making policy in a sequential environment by imitating trajectories collected from an expert demonstrator. Inspired by Generative Adversarial Networks (GANs) [2], GAIL consists of a learned policy serving as a generator, and a discriminator distinguishing expert trajectories from generated ones. The learned policy is optimized through Reinforcement Learning (RL) with a reward signal derived from the discriminator. This paradigm offers distinct advantages over other imitation learning strategies such as Inverse Reinforcement Learning (IRL), which requires an explicit model of the reward function [3], and Behavior Cloning (BC), which suffers from a distribution mismatch during roll-outs [4].

Meanwhile, GAIL does bring certain challenges. One key issue it inherits from GANs is *instability* during training [5]. GAIL presents a difficult minimax optimization problem, where the convergence of the discriminator and the policy generator towards their optimal points is not guaranteed in general. This problem manifests in practice as oscillating training curves and an inconsistency in matching the expert's performance (Fig. 1). However, recent empirical works [6–10] on GAIL mostly focus on improving the sample efficiency and final return of the learned policy, without directly resolving the problem of unstable training. On the other hand, theoretical works [11–14] on GAIL's convergence are based on strong assumptions and do not yield a practical algorithm for stabilizing training.

---

\* Corresponding author.

38th Conference on Neural Information Processing Systems (NeurIPS 2024).

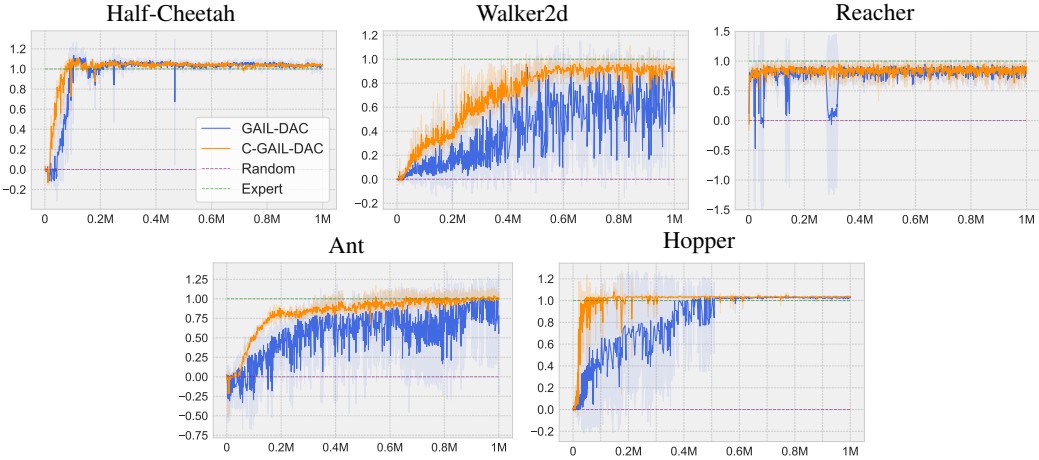

Figure 1: Normalized return curves for controlled GAIL-DAC with four expert demonstrations on five MuJoCo environments averaged over five random seeds. The x-axis represents the number of gradient step updates in millions and the y-axis represents the normalized environment reward, where 1 stands for the expert policy return and 0 stands for the random policy return

In this paper, we study GAIL's training stability using control theory. We describe how the generative policy and discriminator evolve over training at timestep $t$ with a dynamical system of differential equations. We study whether the system can converge to the *desired state* where the generator perfectly matches with the expert policy and the discriminator cannot distinguish generated from expert trajectories. This surprisingly reveals that the desirable state is not an equilibrium of the system, indicating that existing algorithms do not converge to the expert policy, even with unlimited data and model capacity. In response, we study a "one-step GAIL" setting, and design a controller that does create an equilibrium at the desired state. We theoretically prove that this controller achieves *asymptotic stability* around this desired state, which means that if initialized within a sufficiently close radius, the generator and discriminator will indeed converge to it.

Motivated by our theoretical analysis, we propose C-GAIL, which incorporates a pragmatic controller that can be added as a regularization term to the loss function to stabilize training in practice. Empirically we find that our method speeds up the convergence, reduces the range of oscillation in the return curves (shown on Fig. 1), and matches the expert's distribution more closely on GAIL-DAC and other imitation learning methods for a suite of MuJoCo control tasks.

## 1.1 Related Work

**Adversarial imitation learning.** Inspired by GANs and IRL, Adversarial Imitation Learning (AIL) has emerged as a popular technique to learn from demonstrations. GAIL [1] formulated the problem as matching an occupancy measure under the maximum entropy RL framework, with a discriminator providing the policy reward signal, bypassing the need to recover the expert's reward function. Several advancements were subsequently proposed to enhance performance and stability. For instance, AIRL [6] replaced the Shannon-Jensen divergence of GAIL by KL divergence. Baram et al. [15] explored combining GAIL with model-based reinforcement learning. DAC [9] utilized a replay buffer to remove the need for importance sampling and address the issue of absorbing states. Other empirical works such as [7–10, 16] helped improve the sample efficiency and final return of GAIL. In contrast, our work focuses on the orthogonal direction of training stability.

Meanwhile, the convergence behaviors of AIL have also been investigated theoretically. Chen et al. [11] proved that GAIL convergences to a stationary point (not necessarily the desired state). The convergence to the desired state has only been established under strong assumptions such as i.i.d. samples and linear MDP [12, 13] and strongly concave objective functions [14]. However, existing theory has not analyzed the convergence behavior to the desired state in a general setting, and has so far not presented practically useful algorithms to improve GAIL's convergence. Our analysis in Sec. 3.2 show that GAIL actually *cannot* converge to the desired state under general settings.

Additionally, our proposed controller achieves not only a theoretical convergence guarantee, but also empirical improvements in terms of convergence speed and range of oscillation.

**Control theory in GANs.** Control theory has recently emerged as a promising technique for studying the convergence of GANs. Xu et al. [17] designed a linear controller which offers GANs local stability. Luo et al. [18] utilized a Brownian motion controller which was shown to offer GANs global exponential stability. However, for GAIL, the policy generator involves an MDP transition, which results in a much more complicated dynamical system induced by a policy acting in an MDP rather than a static data generating distribution. Prior theoretical analysis and controllers are therefore inapplicable. We adopt different analysis and controlling techniques, to present new stability guarantee, controller, and theoretical results for the different dynamical system of GAIL.

## 2 Preliminaries

We start by formally introducing our problem setting, as well as necessary definitions and theorems relating to the stability of dynamical systems represented by Ordinary Differential Equations (ODEs).

### 2.1 Problem Setting

Consider a Markov Decision Process (MDP), described by the tuple $\langle \mathcal{S}, \mathcal{A}, \mathcal{P}, r, p_0, \gamma \rangle$, where $\mathcal{S}$ is the state space, $\mathcal{A}$ is the action space, $\mathcal{P}(s'|s, a)$ is the transition probability function, $r(s, a)$ is the reward function, $p_0$ is the probability distribution of the initial state $s_0$, and $\gamma \in [0, 1]$ is the discount factor. We work on the $\gamma$-discounted infinite horizon setting, and define the expectation with respect to a policy $\pi \in \Pi$ as the (discounted) expectation over the trajectory it generates. For some arbitrary function $g$ we have $\mathbb{E}_\pi[g(s, a)] \triangleq \mathbb{E}[\Sigma_{n=0}^\infty \gamma^n g(s_n, a_n)]$, where $a_n \sim \pi(a_n|s_n)$, $s_0 \sim p_0$, $s_{n+1} \sim \mathcal{P}(s_{n+1}|s_n, a_n)$. Note that we use $n$ to represent the **environment timestep**, reserving $t$ to denote the **training timestep** of GAIL. For a policy $\pi \in \Pi$, We define its (unnormalized) state occupancy $\rho_\pi(s) = \sum_{n=0}^\infty \gamma^n P(s_n = s|\pi)$. We denote $Q^\pi(s, a) = \mathbb{E}_\pi[\log D(\bar{s}, \bar{a}) + \lambda \log \pi(\bar{a}|\bar{s})|s_0 = s, a_0 = a]$ and the advantage function $A^\pi(s, a) = Q^\pi(s, a) - \mathbb{E}_\pi[Q^\pi(s, a)]$. We assume the setting where we are given a dataset of trajectories $\tau_E$ consisting of state-action tuples, collected from an expert policy $\pi_E$. We assume access to interact in the environment in order to learn a policy $\pi$, but do not make use of any external reward signal (except during evaluation).

### 2.2 Dynamical Systems and Control Theory

In this paper, we consider dynamical systems represented by an ODE of the form

$$\frac{dx(t)}{dt} = f(x(t)), \tag{1}$$

where $x$ represents some property of the system, $t$ refers to the timestep of the system and $f$ is a function. The necessary condition for a solution trajectory $\{x(t)\}_{t \geq 0}$ converging to some steady state value is the existence of an 'equilibrium'.

**Definition 2.1. (Equilibrium)** [19] A point $\bar{x}$ is an *equilibrium* of system (1) if $f(\bar{x}) = 0$. Such an equilibrium is also called a *fixed point*, *critical point*, or *steady state*.

Note that a dynamical system is unable to converge if an equilibrium does not exist. A second important property of dynamical systems is 'stability'. The stability of a dynamical system can be described with Lyapunov stability criteria. More formally, suppose $\{x(t)\}_{t \geq 0}$ is a solution trajectory of the above system (1) with equilibrium $\bar{x}$, we define two types of stability.

**Definition 2.2. (Lyapunov Stability)** [20] System (1) is *Lyapunov Stable* if given any $\epsilon > 0$, there exists a $\delta > 0$ such that whenever $\|x(0) - \bar{x}\| \leq \delta$, we have $\|x(t) - \bar{x}\| < \epsilon$ for $0 \leq t \leq \infty$.

**Definition 2.3. (Asymptotic Stability)** [20] System (1) is *asymptotic stable* if it is Lyapunov stable, and there exists a $\delta > 0$ such that whenever $\|x(0) - \bar{x}\| \leq \delta$, we have $\lim_{t \to \infty} \|x(t) - \bar{x}\| = 0$.

Note that a dynamical system can be Lyapnuov stable but not asymptotic stable. However, every asymptotic stable dynamical system is Lyapnuov stable.

The field of control theory has studied how to drive dynamical systems to desired states. This can be achieved through the addition of a 'controller' to allow influence over the dynamical system's

evolution, for example creating an equilibrium at some desired state, and making the dynamical system stable around it.

**Definition 2.4. (Controller)** [21] A *controller* of a dynamical system is a function $u(t)$ such that

$$\frac{dx(t)}{dt} = f(x(t)) + u(t). \tag{2}$$

The equilibrium and stability criteria introduced for dynamical system (1), equally apply to this controlled dynamical system (2). In order to analyze the stability of a controller $u(t)$ of the controlled dynamical system given an equilibrium $\bar{x}$, the following result will be useful.

**Theorem 2.5.** *(Principle of Linearized Stability) [22] A controlled dynamical system (2) with equilibrium $\bar{x}$ is asymptotically stable if all eigenvalues of $\mathbb{J}(f(\bar{x}) + u(t))$ have negative real parts, where $\mathbb{J}(f(\bar{x}) + u(t))$ represents the Jacobian of $f(x(t)) + u(t)$ evaluated at $\bar{x}$.*

**Corollary 2.6.** *If $\mathbb{J}(f(\bar{x}) + u(t))$ has positive determinant and negative trace, all its eigenvalues have negative real parts and the system is asymptotically stable.*

## 3 Analyzing GAIL as a Dynamical System

In this section, we study the training stability of GAIL through the lens of control theory. We derive the differential equations governing the training process of GAIL, framing it as a dynamical system. Then, we analyze the convergence of GAIL and find that it cannot converge to the desired equilibrium due to the entropy term. For simplicity, we limit the theoretical analysis to the original GAIL[1] among many variants [6–10], while the controller proposed in the next section is general.

### 3.1 GAIL Dynamics

GAIL consists of a learned generative policy $\pi_\theta : \mathcal{S} \to \mathcal{A}$ and a discriminator $D_\omega : \mathcal{S} \times \mathcal{A} \to (0,1)$. The discriminator estimates the probability that an input state-action pair is from the expert policy, rather than the learned policy. GAIL alternatively updates the policy and discriminator parameters, $\theta$ and $\omega$. (The parameter subscripts are subsequently dropped for clarity.) The GAIL objective [1] is $\mathbb{E}_\pi[\log(D(s,a))] + \mathbb{E}_{\pi_E}[\log(1 - D(s,a))] - \lambda H(\pi)$, where $\pi_E$ is the expert demonstrator policy, $\pi$ is the learned policy, and $H(\pi) \equiv \mathbb{E}_\pi[-\log \pi(a|s)]$ is its entropy. Respectively, the objective functions for the discriminator and policy (to be maximized and minimized respectively) are,

$$\begin{aligned} V_D(D,\pi) &= \mathbb{E}_\pi[\log D(s,a)] + \mathbb{E}_{\pi_E}[\log(1 - D(s,a))] \\ V_\pi(D,\pi) &= \mathbb{E}_\pi[\log D(s,a)] - \lambda \mathbb{E}_\pi[-\log \pi(a|s)]. \end{aligned} \tag{3}$$

To describe GAIL as a dynamical system, we express how $\pi$ and $D$ evolve during training. For the analysis to be tractable, we study the training dynamics from a variational perspective, by directly considering the optimization of $\pi$ and $D$ in their respective *function spaces*. This approach has been used in other theoretical deep learning works [2, 5, 23] to avoid complications of the parameter space.

We start by considering optimizing Eq. (3) with functional gradient descent with discrete iterations indexed by $m$: $D_{m+1}(s,a) = D_m(s,a) + \beta \frac{\partial V_D(D_m, \pi_m)}{\partial D_m(s,a)}$, and $\pi_{m+1}(a|s) = \pi_m(a|s) - \beta \frac{\partial V_\pi(D_m, \pi_m)}{\partial \pi_m(a|s)}$, where $\beta$ is the learning rate, $m$ the discrete iteration number, $\frac{\partial V_D(D_m, \pi_m)}{\partial D_m(s,a)}$ (similarly for $\frac{\partial V_\pi(D_m, \pi_m)}{\partial \pi_m(a|s)}$) is the *functional derivative* [24] defined via $\partial V_D(D_m, \pi_m) = \int \frac{\partial V_D(D_m, \pi_m)}{\partial D_m(s,a)} \partial D_m(s,a) \; ds \; da$, which implies the total change in $V_D$ upon variation of function $D_m$ is a linear superposition [25] of the local changes summed over the whole range of $(s,a)$ value pairs.

We then consider the limit $\beta \to 0$, where discrete dynamics become continuous ('gradient flow') $\frac{dD_t(s,a)}{dt} = \frac{\partial V_D(D_t, \pi_t)}{\partial D_t(s,a)}$, and $\frac{d\pi_t(a|s)}{dt} = \frac{-\partial V_\pi(D_t, \pi_t)}{\partial \pi_t(a|s)}$. Formally, we consider the evolution of the discriminator function $D_t : \mathcal{S} \times \mathcal{A} \to \mathbb{R}$ and the policy generator $\pi_t : \mathcal{S} \to \mathcal{A}$ over continuous time $t$ rather than discrete time $m$. We derive the training dynamic of GAIL in the following theorem.

**Theorem 3.1.** *The training dynamic of GAIL takes the form (detailed proof in Appendix Lemma C.2)*

$$\frac{dD_t(s,a)}{dt} = \frac{\rho_{\pi_t}(s)\pi_t(a|s)}{D_t(s,a)} - \frac{\rho_{\pi_E}(s)\pi_E(a|s)}{1 - D_t(s,a)}, \tag{4}$$

$$\frac{d\pi_t(a|s)}{dt} = -\rho_{\pi_t}(s)A^{\pi_t}(s,a). \tag{5}$$

## 3.2 On the Convergence of GAIL

Now, we study the optimization stability of GAIL using the dynamical system Eq. (5). The desirable outcome of the GAIL training process, is for the learned policy to perfectly match the expert policy, and the discriminator to be unable to distinguish between the expert and learned policy.

**Definition 3.2. (Desired state)** We define the desired outcome of the GAIL training process as the discriminator and policy reaching $D_t^*(s, a) = \frac{1}{2}, \pi_t^*(a|s) = \pi_E(a|s)$.

We are interested in understanding whether GAIL converges to the desired state. As discussed in Sec. 2.2, the desired state should be the equilibrium of the dynamical system Eq. (5) for such convergence. According to Def. 2.1, the dynamical system should equal to zero at this point, but we present the following theorem (proved in Proposition C.3 and C.5):

**Theorem 3.3.** *The training dynamics of GAIL does not converge to the desired state, and we have*

$$\frac{dD_t^*(s,a)}{dt} = \frac{\rho_{\pi_t^*}(s)\pi_t^*(a|s)}{D_t^*} - \frac{\rho_{\pi_E}(s)\pi_E(a|s)}{1-D_t^*} = 0, \frac{d\pi_t^*(a|s)}{dt} = -\rho_{\pi_E^*}(s)A^{\pi_E^*}(s,a) \neq 0. \quad (6)$$

Hence, the desired state is not an equilibrium, and GAIL will not converge to it. Since $\frac{d\pi_t^*(a|s)}{dt} \neq 0$, even if the system is forced to the desired state, it will drift away from it. We find that the non-equilibrium result is due to the entropy term $\lambda H(\pi)$, and an equilibrium could be achieved by simply setting $\lambda = 0$ (Corollary C.4). However, the entropy term is essential since it resolves the exploration issue and prevents over-fitting. Therefore, we aim to design a controller that not only keeps the entropy term but also improves the theoretical convergence guarantee.

# 4 Controlled GAIL

Having shown in Section 3 that GAIL does not converge to the desired state, this section considers adding a controller to enable the convergence. We design controllers for both the discriminator and the policy. We show that this controlled system converges to the desired equilibrium and also achieves asymptotic stability in a simplified "one-step" setting.

## 4.1 Controlling the Training Process of GAIL

Establishing the convergence for GAIL is challenging since the occupancy measure $\rho_\pi$ involves an expectation over the states generated by playing the policy $\pi$ for infinite many steps. We simplify the analysis by truncating the trajectory length to one: we only consider the evolution from timestep $n$ to $n+1$. We refer this simplified setting as "one-step GAIL", and the convergence guarantee of our proposed algorithm will be established in this simplified setting. Let $p(s)$ be the probability of the state at $s$ on timestep $n$. The objectives for the discriminator and the policy can then be simplified as,

$$\tilde{V}_D(D, \pi) = \int_a \int_s p(s)\pi(a|s) \log D(s,a) + \pi_E(a|s) \log(1 - D(s,a)) \, ds \, da,$$

$$\tilde{V}_\pi(D, \pi) = \int_a \int_s p(s)\pi(a|s) \log D(s,a) + \lambda p(s)\pi(a|s) \log \pi(a|s) \, ds \, da.$$

The gradient flow dynamical system of these functions is,

$$\frac{dD_t(s,a)}{dt} = \frac{p(s)\pi_t(a|s)}{D_t(s,a)} + \frac{p(s)\pi_E(a|s)}{D_t(s,a) - 1}, \quad (7)$$

$$\frac{d\pi_t(a|s)}{dt} = -p(s)(\log D_t(s,a) + \lambda \log \pi_t(a|s) + \lambda). \quad (8)$$

With this "one-step" simplification, the GAIL dynamics now reveal a clearer structure. For a given $(s, a)$ pair, the change of $D(s, a)$ and $\pi(a|s)$ only depends on $D(s, a), \pi(a|s), p(s)$ and $\pi_E(a|s)$ for the same $(s, a)$ pair, without the need to access function values of other $(s, a)$ pairs. Therefore, we can decompose Eq. (7) & (8), which are ODEs of *functions*, into a series of ODEs of *scalar values*. Each ODE only models the dynamics of two scalar values $(D(s, a), \pi(a|s))$ for a particular $(s, a)$ pair. We will add controller to the scalar ODEs, to asymptotically stabilize their dynamical system. Proving that each scalar ODE is stable suggests that the functional ODE will also be stable. Note that

such decomposition is not possible without the "one-step" simplification, since the evolution of $D$ and $\pi$ for all $(s, a)$ pairs is coupled through $\rho_\pi(s)$ and $A^\pi(s, a)$ in Eq. (5).

Based on the above discussion, we now consider the stability of a system of ODEs for two scalar variables $(D(s, a), \pi(a|s))$. With $s, a$ given, we simplify the notation as $x(t) := D_t(s, a)$, $y(t) := \pi_t(s|a)$, $E := \pi_E(a|s)$, $c := p(s)$, so each scalar ODE can be rewritten as,

$$\frac{dx(t)}{dt} = \frac{cy(t)}{x(t)} + \frac{cE}{x(t) - 1}, \frac{dy(t)}{dt} = -c \log x(t) - c\lambda \log y(t) - c\lambda. \tag{9}$$

We showed earlier that the GAIL dynamic in Eq. (5) does not converge to the desired state. Similarly, neither does our simplified 'one-step' dynamic in Eq. (9) converge to the desired state. We now consider the addition of controllers to push our dynamical system to the desired stated. Specifically, we consider linear negative feedback control [26], which can be applied to a dynamical system to reduce its oscillation. We specify our controlled GAIL system as,

$$\frac{\mathrm{d}x(t)}{\mathrm{d}t} = \frac{cy(t)}{x(t)} + \frac{cE}{x(t) - 1} + u_1(t) \tag{10}$$

$$\frac{\mathrm{d}y(t)}{\mathrm{d}t} = -c \log x(t) - c\lambda \log y(t) - c\lambda + u_2(t), \tag{11}$$

where $u_1(t)$ and $u_2(t)$ are the controllers to be designed for the discriminator and policy respectively. Since the derivative of the discriminator with respect to time evaluated at the desired state (Def. 3.2) already equals zero, the discriminator is already able to reach its desired state. Nevertheless, the discriminator can still benefit from a controller to speed up the rate of convergence – we choose a linear negative feedback controller for $u_1(t)$ to push the discriminator towards its desired state. On the other hand, the derivative of the policy generator evaluated at its desired state in Eq. (9) does not equal zero. Therefore, $u_2(t)$ should be set to make Eq. (11) equal to zero evaluated at the desired state. We have designed it to cancel out all terms in Eq. (9) at this desired state, and also provide feasible hyperparameter values for an asymptotically stable system. Hence, we select $u_1(t)$ and $u_2(t)$ to be the following functions,

$$u_1(t) = -k(x(t) - \frac{1}{2}), \tag{12}$$

$$u_2(t) = c\lambda \log E + c \log \frac{1}{2} + c\lambda + \alpha \frac{y(t)}{E} - \alpha, \tag{13}$$

where $k, \alpha$ are hyperparameters. Intuitively, as $k$ gets larger, the discriminator will be pushed harder towards the optimal value of $1/2$. This means the discriminator would converge at a faster speed but may also have a larger radius of oscillation.

## 4.2 Analyzing the Stability of Controlled GAIL

In this section, we apply Theorem 2.5 to formally prove that the controlled GAIL dynamical system described in Eq. (10) & (11) is *asymptotically stable* (Def. 2.3) and give bounds with $\lambda$, $\alpha$, and $k$.

For simplicity, let us define $z(t) = (x(t), y(t))^\top$, and a function $f$ such that $f(z(t))$ is the vector $\left[\frac{cy(t)}{x(t)} + \frac{cE}{x(t)-1} - k\left(x(t) - \frac{1}{2}\right), c \log \frac{1}{2} + c\lambda \log E - c\lambda \log y(t) - c \log x(t) + \alpha \frac{y(t)}{E} - \alpha\right]^\top$. Therefore, our controlled training dynamic of GAIL in Eq. (10) and Eq. (11) can be transformed to the following vector form

$$d(z(t)) = f(z(t))dt. \tag{14}$$

**Theorem 4.1.** *Let assumption 4.2 hold. The training dynamic of GAIL in Eq. (14) is **asymptotically stable** (proof in Appendix D).*

**Assumption 4.2.** We assume $\alpha, k \in \mathbb{R}, k > 0$, $8c^2\lambda - 8c\alpha - 4c^2 + ck\lambda - k\alpha > 0$, and $\frac{k^2 + 32c(-c\lambda + \alpha)}{32c} < 0$.

**Proof sketch.** The first step in proving asymptotic stability of the system in Eq. (14), is to verify whether our desired state is an equilibrium (Def. 2.1). We substitute the desired state, $z^*(t) = (\frac{1}{2}, E)^\top$, into system (14) and verify that $d(z^*(t)) = f(z^*(t)) = 0$. We then find the linearized system about the desired state $d(z(t)) = \mathbb{J}(f(z^*(t)))z(t)dt$. Under Assumption 4.2, we show that $det(\mathbb{J}(f(z^*(t)))) > 0$ and $trace(\mathbb{J}(f(z^*(t)))) < 0$. Finally we invoke Theorem 2.5 and Corollary 2.6 to conclude that the system in Eq. (14) is asymptotically stable.

---

**Algorithm 1** The C-GAIL algorithm

---
1: **Input:** Expert trajectory $\tau_E$ sampled from $\pi_E$, initial parameters $\theta_0$, and $\phi_0$ for generator and discriminator.
2: **repeat**
3:     Sample trajectory $\tau$ from $\pi_\theta$.
4:     Update discriminator parameters $\phi$ with gradient from,
        $\hat{\mathbb{E}}_\tau \left[ \log D(s,a) - \frac{k}{2} \left( D(s,a) - \frac{1}{2} \right)^2 \right] + \hat{\mathbb{E}}_{\tau_E} \left[ \log(1 - D(s,a)) - \frac{k}{2} \left( D(s,a) - \frac{1}{2} \right)^2 \right]$
5:     Update policy parameters $\theta$ with $V_\pi(D,\pi)$ in Eq. 3
6: **until** Stopping criteria reached

---

## 5 A Practical Method to Stabilize GAIL

In this section, we extend our controller from the "one-step" setting back to the general setting and instantiate our controller as a regularization term on the original GAIL loss function. This results in our proposed variant C-GAIL; a method to stabilize the training process of GAIL.

Since the controllers in Eq. (13) are defined in the dynamical system setting, we need to integrate these with respect to time, in order to recover an objective function that can be practically optimized by a GAIL algorithm. Recalling that $V_D(D,\pi)$ and $V_\pi(D,\pi)$ are the original GAIL loss functions for the discriminator and policy (Eq. (3)), we define $V'_D(D,\pi)$ and $V'_\pi(D,\pi)$ as modified loss functions with the integral of our controller applied, such that

$$V'_D(D,\pi) = V_D(D,\pi) - \mathbb{E}_{\pi,\pi_E} \left[ \frac{k}{2} (D(s,a) - \frac{1}{2})^2 \right],$$

$$V'_\pi(D,\pi) = V_\pi(D,\pi) + \mathbb{E}_{\pi,\pi_E} \left[ \frac{\alpha}{2} \frac{\pi^2(a|s)}{\pi_E(a|s)} + (c \log \frac{1}{2} + c\lambda \log \pi_E(a|s) + c\lambda - \alpha)\pi(a|s) \right].$$

Note that the training dynamics with these loss functions are identical to Eq. (10-13) with guaranteed stability under the 'one-step' setting.

While $V'_D(D,\pi)$ can be computed directly, the inclusion of $\pi_E$, the expert policy, in $V'_\pi(D,\pi)$ is problematic – the very goal of the algorithm is to learn $\pi_E$, and we do not have access to it during training. Hence, in our practical implementations, we only use our modified loss $V'_D(D,\pi)$ to update the discriminator, but use the original unmodified policy objective $V_\pi(D,\pi)$ for the policy. In other words, we only add the controller to the discriminator objective $V_D(D,\pi)$. This approximation has no convergence guarantee, even in the one-step setting. Nevertheless, the control theory-motivated approach effectively stabilizes GAIL in practice, as we shall see in Sec. 6. Intuitively, C-GAIL pushes the discriminator to its equilibrium at a faster speed by introducing a penalty controller centered at $\frac{1}{2}$. With proper selection of the hyperparameter $k$ (ablation study provided in Appendix E), the policy generator is able to train the discriminator at the same pace, leading GAIL's training to converge faster with a smaller range of oscillation, and match the expert distribution more closely.

Our C-GAIL algorithm is listed in Alg. 1. It can be implemented by simply adding a regularization term to the discriminator loss. Hence, our method is also compatible with other variants of GAIL, by straightforwardly incorporating the regularization into their discriminator objective function.

## 6 Evaluation

This section evaluates the benefit of integrating the controller developed in Section 4 with popular variants of GAIL. We test the algorithms on their ability to imitate an expert policy in simulated continuous control problems in MuJoCo [27]. Specifically, we consider applying our controller to two popular GAIL algorithms – both the original 'vanilla' GAIL [1] and also GAIL-DAC [9], a state-of-the-art variant which uses a discriminator-actor-critic (DAC) to improve sample efficiency and reduce the bias of the reward function. Additionally, we include supplementary experiments compared with other GAIL variants such as Jena et al. [7] and Xiao et al. [8] in appendix F.

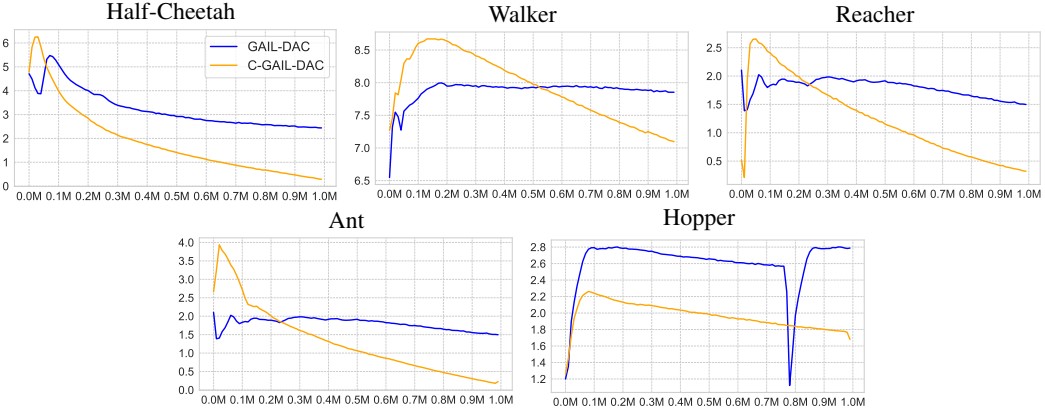

Figure 2: State Wasserstein distance (lower is better) between expert and learned policies, over number of gradient step updates. Our controlled variant matches the expert distribution more closely.

## 6.1 Experimental Setup

We incorporate our controller in vanilla GAIL and GAIL-DAC, naming our controlled variants C-GAIL and C-GAIL-DAC. We leverage the implementations of Gleave et al. [28] (vanilla GAIL) and Kostrikov et al. [9] (GAIL-DAC). Gleave et al. [28] also provide other common imitation learning frameworks – BC, AIRL, and dataset aggregation (DAgger) [29] – which we also compare to.

For C-GAIL-DAC, we test five MuJuCo environments: Half-Cheetah, Ant, Hopper, Reacher and Walker 2D. Our experiments follow the same settings as Kostrikov et al. [9]. The discriminator architecture has a two-layer MLP with 100 hidden units and tanh activations. The networks are optimized using Adam with a learning rate of $10^{-3}$, decayed by $0.5$ every $10^5$ gradient steps. We vary the number of provided expert demonstrations: $\{4, 7, 11, 15, 18\}$, though unless stated we report results using four demonstrations. We assess the normalized return over training for GAIL-DAC and C-GAIL-DAC to evaluate their speed of convergence and stability, reporting the mean and standard deviation over five random seeds. The normalization is done with 0 set to a random policy's return and 1 to the expert policy return.

In addition to recovering the expert's return, we are also interested in how closely our policy generator's and the expert's *state distribution* are matched, for which we use the **state Wasserstein** [30]. This requires samples from two distributions, collected by rolling out the expert and learned policy for 100 trajectories each. We then use the POT library's 'emd2' function [31] to compute the Wasserstein distance, using the L2 cost function with a uniform weighting across samples.

To evaluate C-GAIL, we follow the experimental protocol from Gleave et al. [28], both for GAIL and other imitation learning baselines. These are evaluated on Ant, Hopper, Swimmer, Half-Cheetah and Walker 2D. For C-GAIL, we change only the loss and all other GAIL settings are held constant. We assess performance in terms of the normalized return. We use this set up to ablate the controller strength hyperparameter of C-GAIL (Appendix E), varying $k \in \{0.1, 1, 10\}$ (ablation study of $\alpha$ is not included since our algorithm only involves controller for the discriminator in practice). Our experiments are conducted on a single NVIDIA GeForce GTX TITAN X.

Table 1: Mean and standard deviation for returns of various IL algorithms and environments

|  | Ant | Half Cheetah | Hopper | Swimmer | Walker2d |
|---|---|---|---|---|---|
| Random | $-349 \pm 31$ | $-293 \pm 36$ | $-53 \pm 62$ | $3 \pm 8$ | $-18 \pm 75$ |
| Expert | $2408 \pm 110$ | $3465 \pm 162$ | $2631 \pm 19$ | $298 \pm 1$ | $2631 \pm 112$ |
| Controlled GAIL | $2411 \pm 21$ | $3435 \pm 50$ | $2636 \pm 8$ | $298 \pm 0$ | $2633 \pm 12$ |
| GAIL | $2087 \pm 187$ | $3293 \pm 239$ | $2579 \pm 85$ | $295 \pm 3$ | $2589 \pm 121$ |
| BC | $1937 \pm 227$ | $3465 \pm 151$ | $2830 \pm 265$ | $298 \pm 1$ | $2672 \pm 95$ |
| AIRL | $-121 \pm 28$ | $1837 \pm 218$ | $2536 \pm 142$ | $269 \pm 8$ | $1329 \pm 134$ |
| DAgger | $3027 \pm 187$ | $1693 \pm 74$ | $2751 \pm 11$ | $344 \pm 2$ | $2174 \pm 132$ |

## 6.2 Results

We compare GAIL-DAC to C-GAIL-DAC in Figure 1 (return), 2 (state Wasserstein), and 3 (convergence speed). Figure 1 shows that C-GAIL-DAC speeds up the rate of convergence and reduces the oscillation in the return training curves across all environments. For instance, on Hopper, C-GAIL-DAC converges 5x faster than GAIL-DAC with less oscillations. On Reacher, the return of GAIL-DAC continues to spike even after matching the expert return, but this does not happen with C-GAIL-DAC. On Walker 2D, the return of GAIL-DAC oscillates throughout training, whereas our method achieves a higher return at has reduced the range of oscillation by more than 3 times. For Half-Cheetah, our method converges 2x faster than GAIL-DAC. For Ant environment, C-GAIL-DAC reduces the range of oscillations by around 10x.

In addition to matching the expert's return faster and with more stability, Figure 2 shows that C-GAIL-DAC also more closely matches the expert's state distribution than GAIL-DAC, with the difference persisting even towards the end of training for various numbers of expert trajectories. Toward the end of training, the state Wasserstein for C-GAIL-DAC is more than two times smaller than the state Wasserstein for GAIL-DAC on all five environments.

Figure 3 shows that these improvements hold for differing numbers of provided demonstrations. It plots the number of gradient steps for GAIL-DAC and C-GAIL-DAC to reach $95\%$ of the max-return for vaious numbers of expert demonstrations. Our method is able to converge faster than GAIL-DAC regardless of the number of demonstrations.

**Hyperparameter sensitivity.** We evaluate the sensitivity to the controller's hyperparameter $k$ using vanilla GAIL. Figure 4 (Appendix E) plots normalized returns. For some environments, minor gains can be found by tuning this hyperparameter, though in general for all values tested, the return curves of C-GAIL approach the expert policy's return earlier and with less oscillations than GAIL. This is an important result as it shows that our regularizer can easily be applied by practitioners without the need for a fine-grained hyperparameter sweep.

**Other imitation learning methods.** Table 1 benchmarks C-GAIL against other imitation learning methods, including BC, AIRL, and DAgger, some of which have quite different requirements to the GAIL framework. The table shows that C-GAIL is competitive with many other paradigms, and in consistently offers the lowest variance between runs of any method. Moreover, we include supplementary experiments compared with Jena et al. [7] and Xiao et al. [8] in appendix F.

## 7 Discussion & Conclusion

This work helped understand and address the issue of training instability in GAIL using the lens of control theory. This advances recent findings showing its effectiveness in other adversarial learning frameworks. We formulated GAIL's training as a dynamical system and designed a controller that stabilizes it at the desired state, encouraging convergence to this point. We showed theoretically that our controlled system achieves asymptotic stability under a "one-step" setting. We proposed a

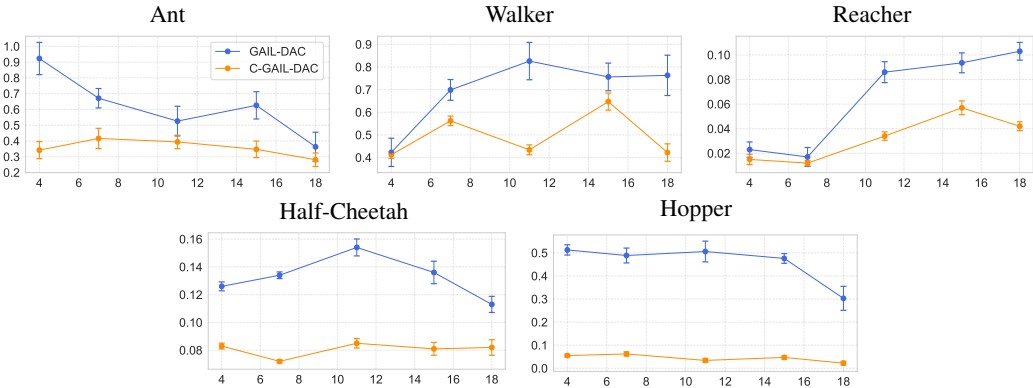

Figure 3: Number of gradient step updates (in millions) required to reach $95\%$ of the max-return for various numbers of expert trajectories on MuJoCo environments averaged over five random seeds.

practical realization of this named C-GAIL, which reaches expert returns both faster and with less oscillation than the uncontrolled variants, and also matches their state distribution more closely.

Whilst our controller theoretically converges to the desired state, and empirically stabilizes training, we recognize several limitations of our work. In our description of GAIL training as a continuous dynamical system, we do not account for the updating of generator and discriminator being discrete as in practice. In our practical implementation of the controller, we only apply the portion of the loss function acting on the discriminator, since the generator portion requires knowing the likelihood of an action under the expert policy (which is precisely what we aim to learn!). We leave it to future work to explore whether estimating the expert policy and incorporating a controller for the policy generator brings benefit.

## Acknowledgments and Disclosure of Funding

We thank Professor Yang Gao for support in discussions. This work was supported by the National Science and Technology Major Project (2021ZD0110502), NSFC Projects (Nos. 62350080, 62376131, 62061136001, 62106123, 62076147, U19A2081, 61972224), and the High Performance Computing Center, Tsinghua University. J.Z is also supported by the XPlorer Prize.

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

## A  Broader impact

This work has provided an algorithmic advancement in imitation learning. As such, we have been cognisant of various issues such as those related to learning from human demonstrators – e.g. privacy issues when collecting data. However, this work avoids such matters by using trained agents as the demonstrators. More broadly, we see our work as a step towards more principled machine learning methods providing more efficient and stable learning, which we believe in general has a positive impact.

## B  Basics of Functional Derivatives

We provide some background on functional derivatives, which are necessary for the derivations in Appendix C. For a more rigorous and detailed introduction, please refer to the Appendix D of [32].

**Definition B.1.** (functionals) We define a *functional* $F[y] : f \longrightarrow \mathbb{R}$ to be an operator that takes a function $y(x)$ and returns a scalar value $F[y]$.

**Definition B.2.** (functionals derivatives) We consider how much a functional $F[y]$ changes when we make a small change $h\eta(x)|_{h \to 0}$ to the function. If we have

$$\int \frac{\partial F}{\partial y}(x)\eta(x)\,\mathrm{d}x = \lim_{h \to 0} \frac{F(y+h\eta) - F(y)}{h} = \frac{dF(y+h\eta)}{dh}\bigg|_{h=0},$$

such that

$$F(y+h\eta) = F(y) + h\int \frac{\partial F}{\partial y}(x)\eta(x)\,\mathrm{d}x + \mathcal{O}(h^2),$$

then $\frac{\partial F}{\partial y}$ is called the *functional derivative* of $F$ with respect to function $y$.

**Theorem B.3.** *(chain rule) If the function $y$ is controlled by some weights $\theta$ can be rewritten as $y_\theta$. We have*

$$\frac{\partial F(y_\theta)}{\partial \theta} = \int \frac{\partial F}{\partial y}(x)\frac{\partial y_\theta}{\partial \theta}(x)\mathrm{d}x.$$

## C  Detailed Theoretical Analysis of General GAIL

$$\max_D V_D(D, \pi) = \mathbb{E}_\pi[\log D(s,a)] + \mathbb{E}_{\pi_E}[\log(1 - D(s,a))] \tag{15}$$

$$\min_\pi V_\pi(D, \pi) = \mathbb{E}_\pi[\log D(s,a)] - \lambda\mathbb{E}_\pi[-\log \pi(a|s)]. \tag{16}$$

**Lemma C.1.** *Given that $\pi_\theta$ is a parameterized policy. Define the training objective for entropy-regularized policy optimization as*

$$J(\theta) = \mathbb{E}_{\pi_\theta}[r(s,a)] - \lambda\mathbb{E}_{\pi_\theta}[-\log \pi_\theta(a|s)].$$

*Its gradient satisfies*

$$\frac{\partial}{\partial \theta}J(\theta) = \mathbb{E}_{\pi_\theta}\Big[\frac{\partial \log \pi_\theta(a|s)}{\partial \theta}Q^{\pi_\theta}(s,a)\Big] = \mathbb{E}_{\pi_\theta}\Big[\frac{\partial \log \pi_\theta(a|s)}{\partial \theta}A_E^{\pi_\theta}(s,a)\Big],$$

*where $Q^{\pi_\theta}(s,a)$ and $A^{\pi_\theta}(s,a)$ are defined as*

$$Q^{\pi_\theta}(s,a) := E_{\pi_\theta}[r(\bar{s},\bar{a}) + \lambda\log\pi_\theta(\bar{a}|\bar{s})|s_0 = s, a_0 = a], \quad A^{\pi_\theta}(s,a) := Q^{\pi_\theta}(s,a) - \mathbb{E}_{\pi_\theta}Q^{\pi_\theta}(s,a).$$

*Proof.*

$$\frac{\partial}{\partial \theta} J(\theta) = \frac{\partial}{\partial \theta} \mathbb{E}_{\pi_\theta}[r(s,a)] - \lambda \mathbb{E}_{\pi_\theta}[-\log \pi_\theta(a|s)]$$

$$= \frac{\partial}{\partial \theta} \int \rho_{\pi_\theta}(s)\pi_\theta(a|s)r(s,a)\mathrm{d}a\mathrm{d}s + \lambda \frac{\partial}{\partial \theta} \int \rho_{\pi_\theta}(s)\pi_\theta(a|s)\log \pi_\theta(a|s)\mathrm{d}a\mathrm{d}s$$

$$= \int \frac{\partial \rho_{\pi_\theta}(s)\pi_\theta(a|s)}{\partial \theta}r(s,a)\mathrm{d}a\mathrm{d}s + \lambda \int \frac{\partial \rho_{\pi_\theta}(s)\pi_\theta(a|s)}{\partial \theta}\log \pi_\theta(a|s)\mathrm{d}a\mathrm{d}s + \lambda \int \rho_{\pi_\theta}(s)\pi_\theta(a|s)\frac{\partial \log \pi_\theta(a|s)}{\partial \theta}\mathrm{d}a\mathrm{d}s$$

$$= \int \frac{\partial \rho_{\pi_\theta}(s)\pi_\theta(a|s)}{\partial \theta}[r(s,a) + \lambda \log \pi_\theta(a|s)]\mathrm{d}a\mathrm{d}s + \lambda \int \rho_{\pi_\theta}(s)\pi_\theta(a|s)\frac{1}{\pi_\theta(a|s)}\frac{\partial \pi_\theta(a|s)}{\partial \theta}\mathrm{d}a\mathrm{d}s$$

$$= \int \frac{\partial \rho_{\pi_\theta}(s)\pi_\theta(a|s)}{\partial \theta}[r(s,a) + \lambda \log \pi_\theta(a|s)]\mathrm{d}a\mathrm{d}s + \lambda \int \rho_{\pi_\theta}(s)\frac{\partial}{\partial \theta}\int \pi_\theta(a|s)\mathrm{d}a\mathrm{d}s$$

$$= \int \frac{\partial \rho_{\pi_\theta}(s)\pi_\theta(a|s)}{\partial \theta}[r(s,a) + \lambda \log \pi_\theta(a|s)]\mathrm{d}a\mathrm{d}s$$

$$= \frac{\partial \mathbb{E}_{\pi_\theta}[r(s,a) + \lambda \log \pi_{\theta'}(a|s)]}{\partial \theta}\Big|_{\theta'=\theta}$$

The above derivation suggests that we can view the entropy term as an additional fixed reward $r'(s,a) = \lambda \log \pi_\theta(a|s)$. Applying the Policy Gradient Theorem, we have

$$\frac{\partial}{\partial \theta} J(\theta) = \mathbb{E}_{\pi_\theta}[\frac{\partial \log \pi_\theta(a|s)}{\partial \theta}Q^{\pi_\theta}(s,a)] = \mathbb{E}_{\pi_\theta}[\frac{\partial \log \pi_\theta(a|s)}{\partial \theta}A^{\pi_\theta}(s,a)],$$

where $Q^{\pi_\theta}$ is similar to the classic Q-function but with an extra "entropy reward" term. $\qquad \square$

**Lemma C.2.** *The functional derivatives for the two optimization objectives*

$$V_D(D,\pi) = \mathbb{E}_\pi[\log D(s,a)] + \mathbb{E}_{\pi_E}[\log(1 - D(s,a))]$$

$$V_\pi(D,\pi) = \mathbb{E}_\pi[\log D(s,a)] - \lambda \mathbb{E}_\pi[-\log \pi(a|s)]$$

*respectively satisfy*

$$\frac{\partial V_D}{\partial D} = \frac{\rho_\pi(s)\pi(a|s)}{D(s,a)} - \frac{\rho_{\pi_E}(s)\pi_E(a|s)}{1 - D(s,a)}.$$

$$\frac{\partial V_\pi}{\partial \pi} = \rho_\pi(s)A^\pi(s,a).$$

*where $A^\pi$ follows the same definition as in Lemma C.1.*

$$Q^\pi(s,a) := E_\pi[\log D(\bar{s},\bar{a}) + \lambda \log \pi(\bar{a}|\bar{s})|s_0 = s, a_0 = a], \quad A^\pi(s,a) := Q^\pi(s,a) - \mathbb{E}_{\pi(a|s)}Q^\pi(s,a).$$

*Proof.* Regarding $\frac{\partial V_D}{\partial D}$, by definition of $\mathbb{E}_\pi$ we have

$$V_D(D,\pi) = \int \rho_\pi(s)\pi(a|s)\log D(s,a)\mathrm{d}a\mathrm{d}s + \int \rho_{\pi_E}(s)\pi_E(a|s)\log(1 - D(s,a))\mathrm{d}a\mathrm{d}s$$

according to the *chain rule* [25] of functional derivative, we have

$$\frac{\partial V_D}{\partial D} = \frac{\rho_\pi(s)\pi(a|s)}{D(s,a)} - \frac{\rho_{\pi_E}(s)\pi_E(a|s)}{1 - D(s,a)}$$

Regarding $\frac{\partial V_\pi}{\partial \pi}$, suppose $\pi$ is parameterized by $\theta$. The chain rule for functional derivative states

$$\frac{\partial V_\pi}{\partial \theta} = \int \frac{\partial V_\pi}{\partial \pi}\frac{\partial \pi}{\partial \theta}\mathrm{d}a\mathrm{d}s.$$

According to Lemma C.1, we have

$$\frac{\partial V_\pi}{\partial \theta} = \mathbb{E}_\pi [\frac{\partial \log \pi(a|s)}{\partial \theta} A^\pi(s,a)]$$

$$= \int \rho_\pi(s) \pi(a|s) \frac{\partial \log \pi(a|s)}{\partial \theta} A^\pi(s,a) \mathrm{d}a \mathrm{d}s$$

$$= \int \rho_\pi(s) \frac{\partial \pi(a|s)}{\partial \theta} A^\pi(s,a) \mathrm{d}a \mathrm{d}s.$$

Therefore, we have

$$\frac{\partial V_\pi}{\partial \pi} = \rho_\pi(s) A^\pi(s,a) = \rho_\pi(s) [Q^\pi(s,a) - \mathbb{E}_\pi Q^\pi(s,a)].$$

$\square$

**Proposition C.3.** *The constrained optimization problem*

$$\min_\pi V_\pi(D,\pi) = \mathbb{E}_\pi[\log D(s,a)] - \lambda \mathbb{E}_\pi[-\log \pi(a|s)] \quad s.t. \int \pi(a|s) = 1$$

*does not converge when* $\pi = \pi_E$ *and* $D(s,a) = \frac{1}{2}$ *for* $\forall s,a$. *Namely,*

$$\frac{\partial V_\pi}{\partial \pi}|_{\pi(s,a)=\pi_E(s,a), D(s,a)=\frac{1}{2}} \neq 0.$$

*When* $\pi = \pi_E$ *and* $D(s,a) = \frac{1}{2}$, *we have*

$$Q^\pi(s,a) = E_{\pi_E}[\lambda \log \pi_E(\bar{a}|\bar{s}) - \log 2 | s_0 = s, a_0 = a]$$

$$= \sum_{n=0}^{\infty} \gamma^n \int p(s_n = \bar{s}|s_0 = s, a_0 = a) \int \pi_E(\bar{a}|\bar{s})[\lambda \log \pi_E(\bar{a}|\bar{s}) - \log 2] \mathrm{d}\bar{a} \mathrm{d}\bar{s}$$

$$= -\sum_{n=0}^{\infty} \gamma^n \int p(s_n = \bar{s}|s_0 = s, a_0 = a)[\lambda H(\pi_E(\cdot|\bar{s})) + \log 2] \mathrm{d}\bar{s}$$

$$A^\pi(s,a) = Q^\pi(s,a) - \mathbb{E}_\pi Q^\pi(s,a)$$

$$= \sum_{n=0}^{\infty} \gamma^n [p(s_n = \bar{s}|s_0 = s) - p(s_n = \bar{s}|s_0 = s, a_0 = a)] \lambda H(\pi_E(\cdot|\bar{s}))$$

*According to Lemma C.2,*

$$\frac{\partial V_\pi}{\partial \pi} = \rho_{\pi_E}(s) A^{\pi_E}(s,a) = \rho_{\pi_E}(s) A^{\pi_E}(s,a)$$

*Recall that we have* $A^{\pi_E}(s,a) = \sum_{n=0}^{\infty} \gamma^n [p_{\pi_E}(s_n = \bar{s}|s_0 = s) - p_{\pi_E}(s_n = \bar{s}|s_0 = s, a_0 = a)] \lambda H(\pi_E(\cdot|\bar{s}))$. *Since* $\pi_E$ *can by any policy distribution determined by the expert dataset,* $H(\pi_E(\cdot|\bar{s}))$ *can be any value for various* $s$ *and* $a$. *Additionally, for different actions* $a_1 \neq a_2$, *we cannot guarantee* $p_{\pi_E}(s_n = \bar{s}|s_0 = s, a_0 = a_1) = p_{\pi_E}(s_n = \bar{s}|s_0 = s, a_0 = a_2)$. *Thus* $\frac{\partial V_\pi}{\partial \pi}$ *is not a constant and relies on action* $a$. $A^{\pi_E}(s,a) = 0$ *cannot hold, and thus* $\frac{\partial V_\pi}{\partial \pi} \neq 0$.

**Corollary C.4.** *This is a corollary of Proposition C.3. When* $\pi(s,a) = \pi_E(s,a)$, $D(s,a) = \frac{1}{2}$, *and the entropy term is excluded from the GAIL objective, we find,* $\frac{\partial V_\pi}{\partial \pi} = 0$.

*Proof.* Exclusion of the entropy term can be achieved by setting $\lambda = 0$. Then we have,

$$\frac{\partial V_\pi}{\partial \pi} = \rho_{\pi_E}(s) A^{\pi_E}(s,a) \tag{17}$$

$$= \rho_{\pi_E}(s) \sum_{n=0}^{\infty} \gamma^n [p(s_n = \bar{s}|s_0 = s) - p(s_n = \bar{s}|s_0 = s, a_0 = a)] \lambda H(\pi_E(\cdot|\bar{s})) \tag{18}$$

$$= 0 \tag{19}$$

$\square$

**Proposition C.5.** *The optimization problem*

$$\max_D V_D(D, \pi) = \mathbb{E}_\pi[\log D(s, a)] + \mathbb{E}_{\pi_E}[\log(1 - D(s, a))]$$

*converges when $\pi = \pi_E$ and $D(s, a) = \frac{1}{2}$ for $\forall s, a$. Namely,*

$$\frac{\partial V_D}{\partial D}\big|_{\pi(s,a)=\pi_E(s,a), D(s,a)=\frac{1}{2}} = 0.$$

*Proof.* According to the chain rule of functional derivative, we have

$$\begin{aligned}
\frac{\partial V_D}{\partial D} &= \frac{\partial \mathbb{E}_\pi[\log D] + \mathbb{E}_{\pi_E}[\log(1 - D)]}{\partial D} \\
&= \mathbb{E}_\pi\left[\frac{1}{D}\right] - \mathbb{E}_{\pi_E}\left[\frac{1}{1 - D}\right] \\
&= \mathbb{E}_{\pi_E}\left[\frac{1}{D} - \frac{1}{1 - D}\right] \\
&= \mathbb{E}_{\pi_E}[2 - 2] \\
&= 0
\end{aligned}$$

$\square$

# D    Proof of Theorem 4.1

**Theorem D.1.** *Let assumption 4.2 holds. The training dynamic of GAIL in Eq. (14) is **asymptotically stable**.*

*Proof.* To analyze the convergence and stability behavior of system 14, first we need to verify definition 2.1 to make sure our goal functions are equilibrium points. Then we apply theorem 2.5 to prove system 14 is asymptotically stable. Notice that $z^*(t) = (\frac{1}{2}, E)\top$, then we substitute this goal function to system 14

$$d(z^*(t)) = f(z^*(t)) = 0$$

We the compute the linearized system near the goal function such that

$$d(z(t)) = \mathbb{J}(f(z^*(t)))z(t)dt, \tag{20}$$

where $\mathbb{J}$ is the Jacobian of function $f$. Therefore,

$$\mathbb{J}(f(z^*(t))) = \begin{pmatrix} -\frac{cy(t)}{x(t)^2} - \frac{cE}{(x(t)-1)^2} - k & \frac{c}{x(t)} \\ -\frac{c}{x(t)} & -\frac{c\lambda}{y(t)} + \frac{\alpha}{E} \end{pmatrix}_{(\frac{1}{2}, E)}, \tag{21}$$

which after evaluation becomes

$$\mathbb{J}(f(z^*(t))) = \begin{pmatrix} -8cE - k & 2c \\ -2c & \frac{-c\lambda + \alpha}{E} \end{pmatrix} \tag{22}$$

Then we compute the determinate and trace of $\mathbb{J}(f(z^*(t)))$, which

$$det(\mathbb{J}(f(z^*(t)))) = \frac{(8c^2\lambda - 8c\alpha - 4c^2)E + (ck\lambda - k\alpha)}{E} \tag{23}$$

$$trace(\mathbb{J}(f(z^*(t)))) = \frac{-8cE^2 - kE - c\lambda + \alpha}{E} \tag{24}$$

Since $E = \pi_E(a|s)$ has range $[0, 1]$, therefore we have $det(\mathbb{J}(f(z^*(t)))) > 0$, if

$$ck\lambda - k\alpha > 0 \tag{25}$$

$$8c^2\lambda - 8c\alpha - 4c^2 + ck\lambda - k\alpha > 0 \tag{26}$$

The graph of $trace(\mathbb{J}((f(z^*(t)))))$ is also a downward hyperbola with middle point $(\frac{-k}{16c}, \frac{k^2+32c(-c\lambda+\alpha)}{32c})$. Therefore, $trace(\mathbb{J}((f(z^*(t))))) < 0$, if

$$\frac{k^2 + 32c(-c\lambda + \alpha)}{32c} < 0. \tag{27}$$

Note that this implies $ck\lambda - k\alpha > 0$, since

$$\frac{k^2 + 32c(-c\lambda + \alpha)}{32c} < 0 \tag{28}$$

$$\frac{k^2}{32c} - c\lambda + \alpha < 0 \tag{29}$$

$$-\frac{k^2}{32c} + c\lambda - \alpha > 0 \tag{30}$$

$$c\lambda - \alpha > \frac{k^2}{32c} > 0. \tag{31}$$

As a result, system 14 is asymptotically stable if assumptions 4.2 hold.

$\square$

# E    Ablation on $k$

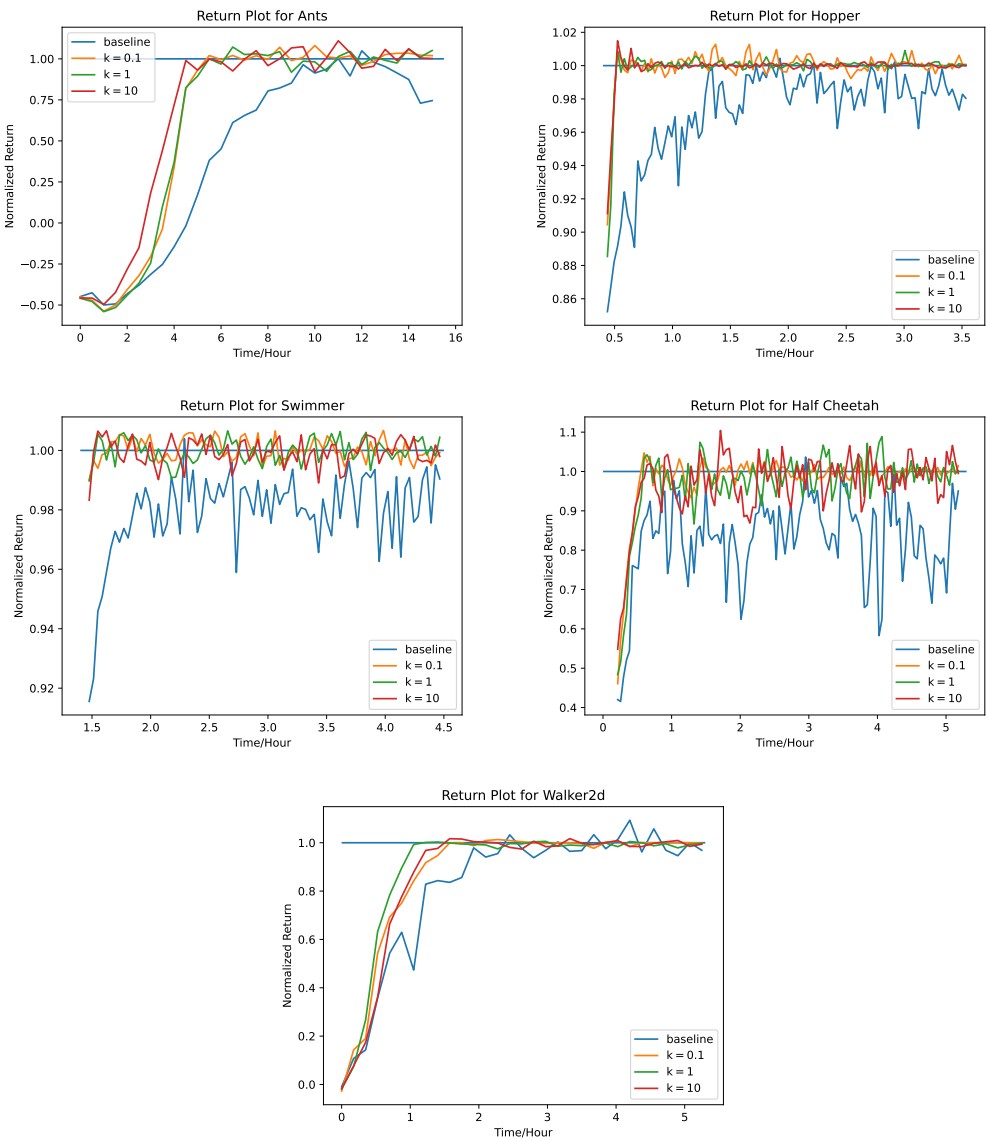

Figure 4: Normalized returns curves for controlled GAIL with $k = 0.1$, $k = 1$, and $k = 10$ on MuJoCo environments, where on the y-axis, 1 represents expert policy return and 0 represents random policy return

# F Comparison with Other IL Methods

|  | BC+GAIL | WAIL | GAIL-DAC | Ours |
|---|---|---|---|---|
| Expert Trajectories | 200 | 4 | 4 | 4 |
| Half-Cheetah | 4558.09 ± 89.50 | 3660.49 ± 217.60 | 5097.51 ± 62.93 | 5102.32 ± 5.80 |
| Hopper | 3554.35 ± 165.7 | 3573.74 ± 12.98 | 3521.14 ± 36.81 | 3586.24 ± 8.78 |
| Reacher | -7.98 ± 2.66 | -9.81 ± 3.23 | -10.56 ± 5.61 | -8.61 ± 1.46 |
| Ant | 3941.69 ± 944.67 | 4173.97 ± 213.46 | 3268.29 ± 1301.12 | 4239.40 ± 42.81 |
| Walker2d | 6799.93 ± 387.85 | 5274.72 ± 981.72 | 4558.99 ± 302.64 | 5912.83 ± 146.58 |

Table 2: Return for each environment on various GAIL algorithms.

|  | BC+GAIL | WAIL | GAIL-DAC | Ours |
|---|---|---|---|---|
| Expert Trajectories | 200 | 4 | 4 | 4 |
| Half-Cheetah | 0.73M | 15.24M | 0.13M | 0.08M |
| Hopper | 30.36M | 74.11M | 0.51M | 0.06M |
| Reacher | 0.68M | 10.84M | 0.02M | 0.01M |
| Ant | 0.52M | 98.50M | 0.92M | 0.34M |
| Walker2d | 1.26M | 79.48M | 0.43M | 0.41M |

Table 3: Number of Iterations needed to reach 95% of return

|  | Expert | DiffAIL | C-DiffAIL |
|---|---|---|---|
| Hopper | 3402 | 3382.03 ± 142.86 | 3388.28 ± 41.23 |
| HalfCheetah | 4463 | 5362.25 ± 96.92 | 4837.34 ± 30.58 |
| Ant | 4228 | 5142.60 ± 90.05 | 4206.64 ± 36.52 |
| Walker2d | 6717 | 6292.28 ± 97.65 | 6343.89 ± 33.67 |

Table 4: Final reward with 1 trajectory in diffusion-based GAIL [2], both vanilla and controlled variant. Mean and standard deviation over five runs

# G Comparison in Atari tasks

|  | Expert (PPO) | GAIL | C-GAIL |
|---|---|---|---|
| BeamRider | $2637.45 \pm 1378.23$ | $1087.60 \pm 559.09$ | $1835.27 \pm 381.84$ |
| Pong | $21.32 \pm 0.0$ | $-1.73 \pm 18.13$ | $0.34 \pm 8.93$ |
| Q*bert | $598.73 \pm 127.07$ | $-7.27 \pm 24.95$ | $428.87 \pm 12.72$ |
| Seaquest | $1840.26 \pm 0.0$ | $1474.04 \pm 201.62$ | $1389.47 \pm 80.24$ |
| Hero | $27814.14 \pm 46.01$ | $13942.51 \pm 67.13$ | $23912.73 \pm 32.69$ |

Table 5: Final reward in five Atari tasks. Mean and standard deviation over ten runs

