# OpenReview forum: "C-GAIL: Stabilizing Generative Adversarial Imitation Learning with Control Theory"
_NeurIPS.cc/2024/Conference — NeurIPS 2024 poster_

### Official Review · Reviewer_9gMV · 2024-06-27

**Soundness:** 3
**Presentation:** 2
**Contribution:** 2
**Rating:** 5
**Confidence:** 3

**Summary:**

This paper theoretically analyzes the training dynamics of GAIL, a widely discussed issue in GANs, pointing out that the original GAIL cannot converge to the desired equilibrium. From a control theory perspective, the paper propose C-GAIL, which can achieve asymptotic stability. The paper demonstrate that C-GAIL, compared to several baseline methods, accelerates convergence, reduces oscillations, and more closely approximates the expert distribution in 5 typical MoJoCo control tasks.

**Strengths:**

- Well-written paper with clear objectives. The authors have the intention to share the code.
- The idea is novel for GAIL, although it draws on concepts from GAN.
- Theoretical analysis is complete.
- The method is simple to implement and can be easily applied to various existing GAIL methods.

**Weaknesses:**

- The theoretical analysis is based on quite a few assumptions, and the final implementation method is only a rough approximation of the theoretical results.
- The experimental section is not comprehensive enough. Given that the method is simple and easy to deploy, can its effectiveness be validated in more GAIL variants and environments?
- There is a lack of comparison with the latest GAIL variant algorithms; only GAIL (2016) and GAIL-DAC (2018) are shown in Figures 1, 2, and 3.
- There is a divergence between the theoretical motivation of the paper and the final algorithm implementation.

**Questions:**

- What are the main differences between C-GAIL and similar methods in GANs? It is recommended to provide a separate discussion.
- The authors claim that the reason for the original GAIL's failure to converge to equilibrium is due to the entropy regularization term. Removing entropy can ensure convergence to equilibrium at the cost of exploration. However, if $V^{\prime}_{\pi}$ is not computable, will it affect the policy's exploration? Compared to directly removing entropy, does C-GAIL show an advantage in exploration?
- The theoretical analysis is based on the original GAIL, which measures JS divergence. For variants like WGAIL, LS-GAIL, and f-GAIL, can convergence and stability still be guaranteed?
- How long is each expert demonstration in the experiments? How were the expert demonstrations obtained?
- In Figure 2, what causes the significant drop in the blue line in the Hopper environment? Were the curves plotted using the averages of multiple experiments?
- $\bar{s}$ and $\bar{a}$ in line 89 are undefined.
- Table 2 lacks the expert reward baseline as a reference.

**Limitations:**

The limitations of the method have been outlined in the discussion.

---

> ### Author Rebuttal · Authors · 2024-08-07
>
> Thank you for your review. We hope our response might provide sufficient evidence that our method is general enough to consider upgrading your score. In particular please note the additional experiments on a new environment, and the three additional variants (two already included in Appendix F, one new in the global rebuttal).
>
> **Q1. From theory to practical settings**
>
> A1. Please see our global review A3.
>
>
> **Q2. More environments**
>
> A2. We have included a set of Atari results under the global rebuttal. Please let us know any specific additional environments you'd like to see.
>
>
> **Q3. Latest GAIL variants**
>
> A3. While the results in the paper's main body indeed focus on GAIL and GAIL-DAC, Appendix F Table 2 already includes comparisons with two other recent GAIL variants -- BC-GAIL and WAIL. We will make these results more prominent in the next version of the paper. In addition, we have included a diffusion variant in the global rebuttal. Please let us know any specific additional variants you'd like to see.
>
>
> **Q4. Differences between C-GAIL and similar methods in GANs?**
>
> A4. Compared with GANs, the policy generator of GAIL involves an MDP transition, which results in a much more complicated dynamical system induced by a policy acting in an MDP rather than a static data generating distribution. Prior theoretical analysis and controllers are therefore inapplicable. We adopt different analysis (different ODEs) and controlling techniques (local linearization), to present a new stability guarantee, controller, and theoretical results for the different dynamical system of GAIL.
>
>
> **Q5. Exploration for entropy term**
>
> A5. Thank you for this comment. We understand that inclusion of the entropy term is critical to allow the generator policy to explore the state space. C-GAIL maintains this entropy term, while additionally adding a controller to stabilize the system. Removing the entropy term directly as you suggest is an interesting proposal, but we expect this would cause the algorithm to collapse to some local minimum without fully exploring the space.
>
>
> **Q6. Theoretical guarantee for other GAIL variants**
>
> A6. Our work presents a methodology to analyse the training process as a dynamical system and design a controller to stabilize its training. To design a controller for other GAIL variants, we would start with their objective functions and take a derivative to get their dynamical system respectively. The only difference is on the design of the generator's controller to ensure local linearization theorem. The controller for the discriminator will remain the same.
>
> In this manner, since our implementation of C-GAIL only adds controllers for the discriminator, our C-GAIL is compatible with WAIL, LSGAIL and f-GAIL. We have provided our result compared with WAIL in appendix F.
>
>
> **Q7. Length of expert demonstration and how obtained**
>
> A7. Our main experiments follow the same settings as GAIL-DAC [1], which in turn followed vanilla GAIL [2] which used 50 timesteps per trajectory. The expert policies were trained with TRPO.
>
>
>
> **Q8. Drop in Hopper Figure 2,  number of random seeds**
>
> A8. This is a good observation. We are not certain why there is a sudden dip in Wasserstein distance for Hopper. It's possible that, because the distance is computed between a low number of expert and generated trajectories, there is noise in our measurement, leading to these unexpected spikes.
>
> Figure 2 is computed by averaging five random seeds. We apologize for the confusion as we did not originally include the error bars. We present the standard deviation as table in global rebuttal Q4 and will modify our figure.
>
>
>
> **Q9. Undefined notations**
>
> A9. Thank you for pointing it out. We will properly define them to avoid confusion. They are substitutes notation for $s$ and $a$ since we have the initial condition $s_0 = s$ and $a_0 = a$.
>
> **Q10. Table 2 missing expert return**
>
> A10. Thank you for noticing this. We will add an extra row for expert return. Since we are conducting on the same Mujuco tasks as Figure 1 and Table 1, the expert return is the same.
>
>
> [1] Ilya Kostrikov, Kumar Krishna Agrawal, Debidatta Dwibedi, Sergey Levine, and Jonathan Tompson. Discriminator-actor-critic: Addressing sample inefficiency and reward bias in adversarial imitation learning. arXiv preprint arXiv:1809.02925, 2018.
>
> [2] Jonathan Ho and Stefano Ermon. Generative adversarial imitation learning. In Advances in Neural Information Processing Systems, pp. 4565–4573, 2016.

---

> > ### Comment · Reviewer_9gMV · 2024-08-10
> > **Thanks for your clarification**
> >
> > Thanks for your rebuttal. I still think that there is a significant difference between the theoretical foundation of the paper and the actual algorithm presented. However, competitive experimental results have alleviated this concern. I decided to improve my score to 5.

---

> > > ### Author Response · Authors · 2024-08-12
> > >
> > > Thank you for your response! We are pleased you have found value in our experimental results.

---

### Official Review · Reviewer_qTQn · 2024-07-07

**Soundness:** 3
**Presentation:** 3
**Contribution:** 3
**Rating:** 7
**Confidence:** 3

**Summary:**

This paper proposes a stabilized version of the Generative Adversarial Imitation Learning (GAIL) through control theory, addressing limitations related to achieving equilibrium and slow convergence. The authors conduct a theoretical analysis using control theory to investigate the properties that influence equilibrium achievement. Empirical validation demonstrates that the proposed modifications improve performance and training stability.

**Strengths:**

- The paper addresses an important problem in the stability of GAIL, which is crucial for the practical implementation of imitation learning techniques.
- The paper provides a theoretical foundation, analyzes the GAIL objective, and demonstrates its limitations and proposed improvements.
- The experimental results validate the theoretical claims, showing improved performance and stability in training.

**Weaknesses:**

- It is unclear how the one-step GAIL, a simplified version, generalizes to the full GAIL framework and impacts policy learning.
- The practical settings used in the experiment do not guarantee convergence, raising questions about the need for evaluating one-step GAIL.
- The paper mentions using a linear negative feedback controller to reduce oscillations but lacks details on its implementation and usage.

**Questions:**

1. How does the simplified one-step GAIL compare to the full GAIL in terms of performance and convergence?
2. Can the authors provide a quantitative analysis of training oscillations shown in Figure 1 and correlate them with training stability? It would be beneficial to quantify these oscillations, as they are a major contribution to the paper. It is mentioned 3 times for Walker2d at line 296, but how is this number computed? Similarly, 10x is mentioned in line 298. Additionally, how do these oscillations indicate training stability, and what is the correlation?
3. How is the controller designed and implemented in the system evaluated? Is there a way to automate the design of controllers u_1 and u_2, and what are the empirical hyperparameters involved?
4. In Figure 2, what is the high distance value on the y-axis for C-GAIL-DAC?  Could the authors explain the initial higher values?
5. The number of demonstrations (four) mentioned (line 273) and used in the paper seems small. Can the authors elaborate on how this number suffices for learning effective policies?

**Limitations:**

Yes. The authors have addressed the limitations.

---

> ### Author Rebuttal · Authors · 2024-08-07
>
> Thank you for your review. We're pleased you have seen value in our new control-theoretic approach to understanding and stabilizing GAIL. We appreciate several nuanced observations you raised, and respond to these below.
>
> **Q1. One-step vs full GAIL**
>
> A1. Theoretically, our controllers work under one-step setting. We follow the insights from one-step setting and extend our controller to full GAIL as C-GAIL.  It is difficult to know how to study the difference between the one-step and full setting analytically -- if able to do this, we could directly design controllers for the full setting! It is an interesting direction for future work. However, we have empirically studied the effectiveness of our C-GAIL under the full setting through our experiments. Also see our response A2 to R jmhJ, which notes similar approximations have been used in popular methods such as TRPO, and the global rebuttal A3.
>
>
> **Q2. Quantifying oscillations**
>
> A2. This is a good point. We have used the term "oscillations" informally in the text, without defining it rigorously. A direct way to interpret this is as variance in the reward curves across runs -- viewing the size of the shaded standard deviation in Figure 1 would be one way to quantify this (or reading off the final standard deviations in Table 2). Another way is to consider the oscillations between successive training time steps -- we do not have a single metric capturing this, but it can be viewed on Figure 1 also.
>
> **Q3. Design and implementation of the controllers**
>
> A3. Note that our controllers $u_1$ and $u_2$ only need to be derived once for a given dynamical system. Automating their design in an arbitrary new dynamical system is an interesting direction for future work, but beyond the scope of this paper. We have chosen to use additive controllers, which are implemented as an addition term to the objective functions (line 238-239). This is summarized in Algorithm 1.  Notice that C-GAIL only introduces one additional hyperparameter $k$ (this is one of the attractions of our method). We provide an ablation study in appendix E, finding that C-GAIL is not particularly sensitive to the choice of $k$.
>
>
> **Q4. High initial value on C-GAIL-DAC**
>
> A4. Thank you for this interesting observation. To reiterate; Figure 2 presents the state Wassertein distance for GAIL-DAC and C-GAIL-DAC. The state Wassertein distance measures the difference between expert and generator distributions at a given point in training. Higher values indicate a larger difference between expert and generator. As noted by the reviewer, in 4/5 environments C-GAIL-DAC briefly has a higher (worse) metric than GAIL-DAC, though in the long run always ends up lower (better).
> We are not certain why this occurs, particularly as the reward curves are almost always strictly better for C-GAIL-DAC (Figure 1). Intuitively, since our controller for the discriminator encourages convergence to $ \frac{1}{2}$, the corresponding generator may have a larger distance to the expert one initially. However, as the discriminator approaches its equilibrium, the generator may become stable too as a response.
>
> **Q5. Number of expert demonstrations**
>
> A5. Thank you for this feedback -- we will add a discussion on this in the next version of the paper. We used four expert demonstration to be consistent with the original GAIL-DAC paper [1]. As we understand, one of the advantages of GAIL is that it requires less expert trajectories than other methods (e.g. BC). It's possible that other environments and GAIL variants may require larger numbers of demonstrations (it also depends on the length of each trajectory). We do provide an ablation in Figure 3 with up to 18 demonstrations.
>
> [1] Ilya Kostrikov, Kumar Krishna Agrawal, Debidatta Dwibedi, Sergey Levine, and Jonathan Tompson. Discriminator-actor-critic: Addressing sample inefficiency and reward bias in adversarial imitation learning. arXiv preprint arXiv:1809.02925, 2018.

---

### Official Review · Reviewer_jmhJ · 2024-07-09

**Soundness:** 3
**Presentation:** 3
**Contribution:** 2
**Rating:** 6
**Confidence:** 4

**Summary:**

The paper formulated training process of GAIL as a dynamic system. From control theory’s view, authors pointed out GAIL would not converge to the desired state where the generator perfectly matches with the expert policy and the discriminator cannot distinguish generated from expert trajectories. Hence, authors proposed a new regularizer to stabilize GAIL.

**Strengths:**

1. I have appreciated the logical framework regarding training process of GAIL as a dynamics system, where more control theory tools could be introduced in.

2. Theoretical analyses are solid, making proposed regularized more convincing.

3. The topic selection is meaningful, revealing the issue of training instability in GAIL. Proposed controller is quite necessary for stabilizing GAIL.

**Weaknesses:**

1. This work showed GAIL cannot converge to the only desired state because of biased entropy term. Is it possible that entropy term introduces subtle bias in but improves stability a lot? The case will make your work meaningless. I think more experiment should be interpreted from this view.

2. The GAIL actually minimizes JS divergence between $\rho_{\pi_{E}}$ and $\rho_{\pi}$. For simplicity, the proposed 'C-GAIL' substituted $\rho_{\pi}$ with $\rho$. Does it ignore differences on state distribution? Why the performance still be good?

3. Did this work just substitute a biased regularizer with an unbiased one? It seems that authors should refer success of experiments only to strong convexity of applied controller. I think more details should be supplemented for making your analysis meaningful.

4. More advanced benchmarks should be compared in the experimental section.

**Questions:**

1. Is there a missing parenthesis in the formula at line 181? $\tilde{V}_D(D, \pi)=\iint_0 p(s) \pi(a \mid s) \log D(s, a)+\pi_E(a \mid s) \log (1-D(s, a)) d s d a,$

2. It seems tricky that theoretical analysis is done under a strict condition, where one-time step environment is considered and controllers are applied both on policy and discriminator, but original policy objective is applied for updating in              implementation, although experimental results are good. Could you explain it more, making it convincing?

**Limitations:**

The authors have discussed the limitations and future work in Section 7.

---

> ### Author Rebuttal · Authors · 2024-08-07
>
> Thank you for your review, we are pleased to have communicated the value of our control-theoretic approach. We hope our clarification around the biased-ness of GAIL, and addition of further experiments, might encourage an increase your evaluation of the paper.
>
> **Q1. The entropy term and bias**
>
> A1. We don't believe it's correct to say "the entropy term introduces subtle bias in but improves stability". The role of the entropy term is to allow the generator to explore the state-action space and discover how to reproduce the expert trajectories. But it does not improve stability directly. We have proved that the entropy term prevents the system from converging to the desired global equilibrium, it is not clear to us that there is some other "biased" equilibrium that it is pushed to instead -- empirical evidence comes from the oscillations in the GAIL-DAC training curves (Figure 1) suggesting there is no alternative more stable local equilibrium.
>
>
> **Q2. Ignoring differences on state distribution and performance**
>
> A2. We indeed ignore the effect of policy changes to the state distribution (switching $\rho_{\pi_E}$ with $\rho$) and only consider the influence of $\pi$ on actions in our one-step setting. We believe this is not uncommon in RL theory to simplify analysis. One example is TRPO [1], which similarly allows the original policy distribution to approximate the updated distribution and provides a theoretical justification for this approximation (Eq2 to Eq6 in its paper).
>
> Notice that the methodology of starting with simplified setting in theory and extend it to general setting in practice is also commonly used in GANs (global rebuttal Q3).
>
> Our intuition for why performance is still good with only a controller on the discriminator is that C-GAIL pushes the discriminator to its equilibrium at a faster speed by introducing a penalty controller centered at $\frac{1}{2}$. This leads GAIL’s training to converge faster with a smaller range of oscillation, and match the expert distribution more closely.
>
>
>
>
> **Q3. Biased regularizer and strong convexity**
>
> A3. We do not believe "bias" is the right term here. In control theory we aim for certain equilibria. As per our above response, it's not clear that GAIL with entropy converges to any equilibrium. Further more, we are not replacing this entropy term, but introducing an additional regularizer that does make the system stable around the desired equilibrium.
>
> We are unsure what the convexity comment refers to. None of our results rely on this assumption.
>
> **Q4 More benchmarks**
>
> A4. Thank you for this suggestion. We have added additional experiments as per the global rebuttal.
>
> **Q5. Missing parenthesis**
>
> A5. Thank you for pointing it out. We will modify it.
>
>
> **Q6. One-step setting and lack of policy regularizer**
>
> A6. Please see our response A2 and global rebuttal A3.
>
>
> [1] John Schulman, Sergey Levine, Pieter Abbeel, Michael Jordan, and Philipp Moritz. Trust region policy optimization. ICML, 2015.

---

> > ### Comment · Reviewer_jmhJ · 2024-08-07
> >
> > Authors -- I appreciate your thorough response and am inclined to increase my score from 5 to 6.

---

> > > ### Author Response · Authors · 2024-08-09
> > >
> > > Thank you for your time. We are glad that you find our response helpful.

---

### Official Review · Reviewer_DQNe · 2024-07-10

**Soundness:** 2
**Presentation:** 2
**Contribution:** 2
**Rating:** 5
**Confidence:** 3

**Summary:**

The paper titled "C-GAIL: Stabilizing Generative Adversarial Imitation Learning with Control Theory" addresses the challenge of training instability in Generative Adversarial Imitation Learning (GAIL), a method used to train a generative policy to imitate a demonstrator's behavior. The authors analyze GAIL's optimization from a control-theoretic perspective and identify that GAIL does not converge to the desired equilibrium due to the entropy term in its objective. To resolve this, they propose Controlled-GAIL (C-GAIL), which introduces a differentiable regularization term to the GAIL objective to stabilize training. Empirical results demonstrate that C-GAIL improves upon existing GAIL methods, including GAIL-DAC, by accelerating convergence, reducing oscillation, and more closely matching the expert's policy distribution. The paper contributes a novel control-theoretic approach to stabilize adversarial training in imitation learning, offering both theoretical insights and practical algorithmic advancements.

**Strengths:**

1. The paper introduces a novel application of control theory to stabilize the training process of Generative Adversarial Imitation Learning (GAIL). By analyzing GAIL from a control-theoretic perspective, the authors provide a deeper understanding of the optimization challenges inherent in GAIL and propose a theoretical solution that ensures asymptotic stability.

2. The authors develop Controlled-GAIL (C-GAIL), a practical algorithm that incorporates a regularization term derived from control theory. This algorithm is shown to improve the stability and convergence of GAIL in empirical tests, offering a tangible advancement that can be applied to existing GAIL methods to enhance their performance.

3. The paper provides extensive empirical evidence to support the effectiveness of C-GAIL. Through experiments on MuJoCo control tasks, the authors demonstrate that C-GAIL achieves faster convergence, reduces the range of oscillation in training, and more closely matches the expert's policy distribution compared to both the original GAIL and other variants, showcasing the robustness and practical applicability of their approach.

**Weaknesses:**

1. While the paper proposes a theoretically sound controller for stabilizing GAIL training, the practical implementation of the controller is only applied to the discriminator's objective function. The paper acknowledges that the generator's portion of the controller, which would require knowledge of the expert policy, is not used during training. This limitation means the full potential of the control-theoretic approach is not realized in practice.

2. The paper formulates GAIL training as a continuous dynamical system for the purpose of stability analysis. However, in actual practice, the updates to the generator and discriminator are discrete. The discrepancy between the theoretical model and the practical implementation could potentially affect the real-world applicability of the proposed controller.

3. The stability guarantees provided by the paper rely on certain assumptions, such as specific conditions on the hyperparameters of the controller. These assumptions might not hold in more general settings or across different problem domains, which could limit the broad applicability of the results. Additionally, the paper does not fully explore how violations of these assumptions might impact the performance of C-GAIL.

**Questions:**

1. While the paper demonstrates the effectiveness of C-GAIL in the context of MuJoCo control tasks, how well does the proposed control-theoretic approach generalize to other domains, such as autonomous driving or game playing, where the dynamics of the environment and the complexity of the tasks might be significantly different?

2. Given that the practical updates in GAIL training are discrete whereas the theoretical model assumes a continuous dynamical system, what is the impact of this discrepancy on the long-term stability and performance of the learned policies, especially in tasks with high stochasticity or where the environment reacts non-linearly to actions?

3. The paper mentions that a proper selection of the hyperparameter k is crucial for the effectiveness of the C-GAIL controller. Can the authors provide more insights on how to determine the optimal values for these hyperparameters in a data-driven manner, without relying on extensive hyperparameter tuning, and how sensitive is the performance of C-GAIL to these choices?

**Limitations:**

1. The paper formulates GAIL as a continuous dynamical system for the purpose of stability analysis, but in practice, updates to the generator and discriminator are discrete. This discrepancy between the theoretical model and practical implementation may impact the applicability of the theoretical results in real-world scenarios.

2. The practical implementation of the controller is applied only to the discriminator's objective function. The paper does not provide a method for incorporating a controller for the policy generator, which would require knowledge of the expert policy that is not available during training.

3. The stability guarantees are based on certain assumptions, such as hyperparameter conditions. The paper does not fully explore how the results might be affected when these assumptions do not hold, which could limit the generalizability of the findings to different settings or problem domains.

---

> ### Author Rebuttal · Authors · 2024-08-07
>
> Thank you for taking the time to review our paper. We are pleased we were able to communicate the strengths of the work effectively. We value your feedback on the gap between theory and practice. We have included a general response to this point in the global rebuttal, and here offer more targeted comments. We hope this, along with further experiments in game playing environments, and clarifying the importance of hyperparameter selection, might warrant an uplift in your score.
>
> **Q1 Theoretical controller to practice**
>
> A1. As noted by the reviewer, we were unable to implement the derived generator regularizer since we don't have access to the expert policy. We would like to emphasize that the *theoretical contribution* made by our work remains -- our theory is non-trivial, offering a new way to study GAIL, and insight into what an ideal generator regularizer would look like, which may inspire future work. We showed empirically that the discriminator alone is still of significant practical benefit by itself.
>
> **Q2. Continuous vs Discrete system**
>
> A2. Gradient flow [1] is a widely used technique across optimization theory to transform discrete gradient step updates to a continuous differential equation. This is critical to allow systems to be studied analytically. We agree with the reviewer that understanding what is lost in this conversion is of interest. However, given the ubiquity of the technique we feel it is beyond the scope of our work to study this in depth in our specific setting. We do note that our empirical results, which use discrete updates (algorithm 1), show the benefit of our continuous study does transfer to the discrete setting.
>
>
> **Q3. Hyperparameter assumption requirements**
>
> A3. Assumption 4.2 specifies an allowed range of hyperparameters in order for convergence to be guaranteed in theory. It is not clear how impactful this is when choosing $k$ at implementation time, since one needs to know $c$ ($p(s)$) for each state. So the allowed bounds are more valuable from a theoretical perspective.
>
> **Q4. Experiments on other tasks**
>
> A4. Thank you for this suggestion. We have added game environments (Atari) under the global rebuttal.
>
> **Q5 Selection of hyperparameters**
>
> A5. Thank you for this comment, which suggests we have not communicated the effect of the hyperparameter $k$ clearly. We provided an ablation study on $k$ in Figure 4. This shows that C-GAIL is effective under a wide range of $k$ values, from 0.1 to 10, with only marginal gains to be found by tuning it. As such, one of the benefits of our method is the insensitivity to $k$ and lack of need for tuning. We will emphasize this point more clearly in the next version.
>
> [1] Conley, C., 1988. The gradient structure of a flow: I. Ergodic Theory Dynam. Systems, 8.

---

### Official Review · Reviewer_PPsU · 2024-07-11

**Soundness:** 3
**Presentation:** 3
**Contribution:** 3
**Rating:** 8
**Confidence:** 5

**Summary:**

This paper addresses the problem of unstable training of Generative Adversarial Imitation Learning (GAIL). To this end, the paper studies the convergence of GAIL from a control-theoretic perspective and proposes to employ a regularization term for the discriminator loss function, which can stabilize the training. The experiments in the locomotion domain, e.g., Half-Cheetah and Walker2D, show that the proposed method leads to learned policies that match experts better, achieve better performance, and converge faster. I believe this work provides insightful analyses, presents a promising method, and sufficiently evaluates the proposed method. Hence, I recommend accepting this paper.

**Strengths:**

**Motivation and novelty**
- The motivation for stabilizing GAIL training is convincing.
- Studying this from a control-theoretic perspective is novel to the best of my knowledge.

**Clarity**
- The overall writing is clear.

**Experimental results**
- The experiments in the locomotion domain, e.g., Half-Cheetah and Walker2D, show that the proposed method leads to learned policies that match experts better, achieve better performance, and converge faster.

**Weaknesses:**

**Figure 2 standard deviation**
- Are the results reported in Figure 2 aggregated from five random seeds? Since Figure 2 does not show standard deviation, I am unsure if the gaps are statistically significant.

**Table 1 Hopper results**
- I am wondering why BC (2830) outperforms the expert (2631) in Hopper.

**Limited domains for evaluation**
- The evaluation is limited to locomotion, e.g., Half-Cheetah and Walker2D. Experimenting with the proposed method and the baselines in other domains, such as navigation (grid world or point maze in D4RL), robot arm manipulation (OpenAI Fetch or Shadow Dexterous Hand), and games (Atari) would significantly strengthen the results.

**Visualized state distributions**
- I appreciate the authors showing the state Wasserstein distance between expert and learned policies in Figure 2. I feel it would be informative to visualize the state distributions of expert and learned policies. For example, we can use a grid world navigation task with discrete state and action spaces. Then, we can visualize the state distributions of expert and learned policies as heatmaps and put them side by side for comparison. I believe this would make the claim that C-GAIL-DAC can match the expert state distributions better and more convincing.

**Related work**
- Including the descriptions of more recent IL methods could make the related work more comprehensive, such as
    - Diffusion policy (https://arxiv.org/abs/2303.04137v4 https://arxiv.org/abs/2403.03954)
    - Consistency Policy (https://arxiv.org/abs/2405.07503)
    - Diffusion BC (https://arxiv.org/abs/2302.13335)
    - DiffAIL (https://arxiv.org/abs/2312.06348) / DRAIL (https://arxiv.org/abs/2405.16194v1)

**Questions:**

See above

**Limitations:**

Yes

---

> ### Author Rebuttal · Authors · 2024-08-07
>
> Thank you for your positive review. We are delighted that we were able to communicate the value of our work. We address your questions below, and have provided several new results in the global rebuttal.
>
> **Q1. Figure 2 standard deviation**
>
> A1. The results in Figure 2 are indeed aggregated from five random seeds. Thank you for pointing out that we did not report error bars on the plot. We present the standard deviation as tables in global rebuttal Q4 and will modify our figure.
>
> **Q2. Table 1 Hopper results exceed expert**
>
> A2. This is a good observation, and we do not have a comprehensive answer for why the agent average slightly surpasses expert performance. This also happened in the new DiffAIL results. It's possible that the agent overfits to one of the more successful demonstration trajectories. However, we notice that the expert return is within the error bars of our method.
>
>
> **Q3. Additional experiment domains**
>
> A3. We agree that our results could be more impactful with further domains. We have added in additional environment on Atari as per the global rebuttal.
>
>
> **Q4. Additional IL algorithms in related work**
>
> A4. Thank you for pointing out this gap in our related work, which we will add to the next version of our paper. Moreover, inspired by the reviewer's comment, we have conducted new experiments on DiffAIL in the global rebuttal.

---

> > ### Comment · Reviewer_PPsU · 2024-08-09
> > **Re: Rebuttal by Authors**
> >
> > I appreciate the authors for reporting the error bars, and providing additional Atari experiments on and comparisons to DiffAIL. I believe this paper presents solid contributions and should be accepted. Hence, I increased my score to 8 (strong accept).

---

> > > ### Author Response · Authors · 2024-08-12
> > >
> > > We are pleased we have been able to further strengthen the paper by incorporating your feedback. Thank you again for your time.

---

### Author Rebuttal · Authors · 2024-08-06

# Global Rebuttal for common questions

Thanks to all reviewers for their constructive feedback. We were pleased our work received a favorable assessment. Whilst we address each reviewer's questions individually, this global rebuttal summarizes our response to common points highlighted by several reviewers. 1) Whether C-GAIL's improved stability holds in further environments. 2) How our controlled variant interacts with other SOTA AIL methods. 3) Clarifying the gap between our theory and practical implementation. 4) We computed standard deviations from Figure 2.

**Q1. Additional environments**

A1.
As noted by R PPsU, DQNe & 9gMV, our experiments focused on continuous control in Mujoco tasks. To explore whether C-GAIL brings benefit in other environments we have run additional experiments using (vanilla) GAIL in several Atari games. C-GAIL achieves higher final reward on 4/5 games, always with lower variance.


|| BeamRider | Pong | Q*bert | Seaquest| Hero
|---| -------- | ------- |------- |------- |------- |
Expert (PPO)| 2637.45 $\pm$ 1378.23| 21.32 $\pm$ 0.0| 598.73 $\pm$ 127.07 | 1840.26 $\pm$ 0.0| 27814.14 $\pm$ 46.01|
GAIL |1087.60 $\pm$ 559.09	|-1.73 $\pm$ 18.13| -7.27 $\pm$ 24.95| 1474.04 $\pm$ 201.62| 13942.51 $\pm$ 67.13|
C-GAIL | 1835.27 $\pm$ 381.84 | 0.34 $\pm$ 8.93 | 428.87 $\pm$ 12.72 | 1389.47 $\pm$ 80.24| 23912.73 $\pm$ 32.69

Table R1. Final reward in five Atari tasks [1]. Mean and standard deviation over ten runs. Based on the implementation here: https://github.com/yunke-wang/gail_atari.


**Q2. Additional GAIL variants**

A2.
R 9gMV requested a comparison with more GAIL variants. R PPsU brought our attention to a new family of diffusion-based GAIL algorithms. We were curious to investigate whether the C-GAIL regularizer would bring benefit in this new class of GAIL methods. Run in Mujoco, the table below presents results. Whilst the final performance for both the controlled and non-controlled variants all roughly match (or exceed) the expert return, the standard deviation of C-DiffAIL is consistently smaller than DiffAIL.

|| Hopper | HalfCheetah | Ant | Walker2d|
|---| -------- | ------- |------- |------- |
Expert| 3402| 4463| 4228| 6717|
DiffAIL | 3382.03 $\pm$ 142.86 | 5362.25 $\pm$ 96.92| 5142.60 $\pm$ 90.05 | 6292.28 $\pm$ 97.65|
C-DiffAIL| 3388.28 $\pm$ 41.23 | 4837.34 $\pm$ 30.58 | 4206.64 $\pm$ 36.52 | 6343.89 $\pm$ 33.67

Table R2. Final reward with 1 trajectory in diffusion-based GAIL [2], both vanilla and controlled variant. Mean and standard deviation over five runs. Based on the implementation here: https://github.com/ML-Group-SDU/DiffAIL.




**Q3. From theory to practice**

A3.
Several reviewers correctly noted differences between the assumptions made in our theory, and the practical implementation of our algorithm. These include moving from a continuous flow to discrete updates (R DQNe), studying the one-step setting (R qTQn), only implementing the discriminator regularizer (R DQNe). (R 9gMV also notes this more generally.)

While we provide a specific responses to each assumption separately in our individual responses, we wanted to make a more general response here. We'd like to emphasize that theoretical studies of deep learning usually make simplifying assumptions in order to make progress analytically. This approach of beginning from simplified theory and applying insights it to a practical setting has often been found to be valuable. For example, approaches to stabilize GANs often follow this approach [3][4][5].

**Q4. Figure 2 error bars**

A4. Figure 2 is computed by averaging five random seeds. We did not include error bars in the original version. We present the standard deviation as table below and will modify our figure in the next version. Note that the standard deviations of C-GAIL-DAC are nearly always lower than for GAIL-DAC.

|              | 0.2M | 0.4M | 0.6M | 0.8M | 1M |
|--------------|------|------|------|------|----|
| Half-Cheetah | 1.32|    0.95  |   0.83 | 0. 86| 0.81 |
| Walker       |  0.65  |   0.42 | 0.46  | 0.41  | 0.43 |
| Reacher      | 0.53   |  0.46   | 0.53|  0.55 | 0.52   |
| Ant          |  0.87   | 0.75  | 0.79 | 0.76 | 0.82   |
| Hopper       |   0.37   | 0.29 | 0.34  | 0.67  | 0.46   |
Table R3. Standard deviation for the state Wasserstein distance of GAIL-DAC

|              | 0.2M | 0.4M | 0.6M | 0.8M | 1M |
|--------------|------|------|------|------|----|
| Half-Cheetah | 0.77 |  0.49|  0.44 | 0.36   | 0.38 |
| Walker       | 0.52  | 0.37   | 0.36   | 0.35   | 0.36  |
| Reacher      | 0.61 | 0.41   | 0.37  | 0.39   |0.38    |
| Ant          | 0.65 | 0.62    | 0.58    |0.58  | 0.56   |
| Hopper       |  0.23  | 0.20  | 0.16  | 0.14   | 0.15   |
Table R4. Standard deviation for the state Wasserstein distance of C-GAIL-DAC


[1] Wang et al., 2021. Learning to weight imperfect demonstrations. ICML.

[2] Wang et al., 2024. DiffAIL: Diffusion Adversarial Imitation Learning. AAAI.

[3] Mescheder et al., 2018. Which training methods for GANs do actually converge? ICML.

[4] Xu et al., 2020. Understanding and stabilizing GANs’ training dynamics using control theory. ICML.

[5] Luo et al., 2023. Stabilizing GANs’ training with brownian motion controller. ICML.

---

### Decision · Program_Chairs · 2024-09-25

**Decision:**

Accept (poster)

**Comment:**

This paper makes contributions to stabilize GAIL (generative adversarial imitation learning) via control theory. Four reviewers are all supportive on the paper and two of them are recommending clear acceptance. Through the author-reviewer discussion, several concerns of the reviewers are addressed and the assessment improved.

Strengths of the paper include solid theoretical contributions by connecting control theory with GAIL and novel regularization methods to stabilize GAIL with rich empirical verifications. There are also weaknesses mentioned by the reviewers and many of them are resolved with additional experiments and explanation, including standard deviations in figure 2 and more testing environments.

Therefore, this paper is recommended for acceptance. The authors are also encouraged to discuss (and incorporate the posted discussion on) the simplified assumptions and the generalizability of the theories.